# Comparison of a New Inertial Sensor Based System with an Optoelectronic Motion Capture System for Motion Analysis of Healthy Human Wrist Joints

**DOI:** 10.3390/s19235297

**Published:** 2019-12-01

**Authors:** Michael Alexander Wirth, Gabriella Fischer, Jorge Verdú, Lisa Reissner, Simone Balocco, Maurizio Calcagni

**Affiliations:** 1Division of Plastic Surgery and Hand Surgery, University Hospital Zurich, University of Zurich, Raemistrasse 100, 8091 Zurich, Switzerland; lisa.reissner@usz.ch (L.R.); maurizio.calcagni@usz.ch (M.C.); 2Institute for Biomechanics, ETH Zurich, Raemistrasse 101, 8092 Zurich, Switzerland; 3Dept. Matematics and Informatics, University of Barcelona, Gran Via 585, 08007 Barcelona, Spain; jorgeverdua@gmail.com (J.V.); balocco.simone@gmail.com (S.B.); 4Department of Orthopedics, Balgrist University Hospital, Forchstrasse 340, 8008 Zurich, Switzerland; 5Computer Vision Center, 08193 Bellaterra, Spain

**Keywords:** inertial measurement units, kinematics, motion analysis, optoelectronic motion capture, wrist, range of motion, clinical assessments

## Abstract

This study aims to compare a new inertial measurement unit based system with the highly accurate but complex laboratory gold standard, an optoelectronic motion capture system. Inertial measurement units are sensors based on accelerometers, gyroscopes, and/or magnetometers. Ten healthy subjects were recorded while performing flexion-extension and radial-ulnar deviation movements of their right wrist using inertial sensors and skin markers. Maximum range of motion during these trials and mean absolute difference between the systems were calculated. A difference of 10° ± 5° for flexion-extension and 2° ± 1° for radial-ulnar deviation was found between the two systems with absolute range of motion values of 126° and 50° in the respective axes. A Wilcoxon rank sum test resulted in a no statistical differences between the systems with *p*-values of 0.24 and 0.62. The observed results are even more precise than reports from previous studies, where differences between 14° and 27° for flexion-extension and differences between 6° and 17° for radial-ulnar deviation were found. Effortless and fast applicability, good precision, and low inter-observer variability make inertial measurement unit based systems applicable to clinical settings.

## 1. Introduction

Measurement of wrist kinematics is used in the field of rehabilitation, medicine, and ergonomics. The purpose of motion analysis is to quantify both normal and pathological movements, quantify the degree of impairment, and to assess the change in a patient’s health status during the rehabilitation process [1]. In addition, such measures can assist clinicians in planning rehabilitation strategies to be used to evaluate the effect of applied interventions. However, there is a large discrepancy between the motion analysis systems currently available in the laboratory and those used in clinical practice [1]. In clinical situations, joint range of motion (ROM) measurements are mainly performed with goniometers. This method has been in use since shortly after the first world war [2,3]. Its measurement quality is dependent on the raters abilities and it is restricted to a single plane and to static positions only [4]. Goniometer measurements are susceptible to considerable intra-observer variation, ranging from 5.2–8°, as well as inter-observer variation between 6–10° for the wrist [5]. They also show minimal detectable differences of 11° for wrist flexion and 8° for wrist extension [6]. Another study found even higher minimal detectable differences for goniometer measurements of wrist flexion-extension and radial-ulnar deviation movements of 18° each [4].

Optoelectronic motion capture systems (OMC) are often considered the laboratory gold standard for motion analysis of human joint kinematics [1] due to their high measurement accuracy of 0.1 mm in position [7]. However, most surgeons and hand rehabilitation specialists do not even have access to a specialized laboratory equipped with such a system. Furthermore, these systems are complex to use and time-consuming in post-processing, which is why they are not implemented in everyday clinical use [1]. Yet highly accurate systems only benefit patients if they are actually being used, which is why investigating usability criteria is also important for every new system aimed for clinical use.

Inertial measurement units (IMU) used in a biomechanical context are either based on accelerometers alone, a combination with gyroscopes or a combination with both gyroscopes and magnetometers. If a combination of sensors is used, the advantages of each of these singular electromechanical sensors compensates for the limitations of the others [8]. Accelerometers provide only information on inclination but not on orientation of the IMU. To compensate for this, gyroscopes provide orientation using integration, but these usually suffer from a drift, which can be reduced by magnetometer measurements [9]. In contrast to goniometers, IMU enable dynamic motion to be measured in all three spatial dimensions. Recent developments in commercially available systems and software allow analysis and display of kinematic data in real-time without requiring in-depth knowledge of motion analysis for their application. IMU based systems have been proposed as a cost-effective and easy-to-use alternative to OMC systems [1]. Furthermore, they have been successfully used for motion analysis of various joints of the upper extremity [10,11,12], the lower extremity [13,14] and the whole body with complex tasks [15,16]. A high reproducibility of IMU based measurements of 5–10° and a low inter-observer variability has been reported in the literature [11]. They have therefore been proposed to bridge the gap between clinical and research methods. Possible proposed clinical applications include monitoring of performance during rehabilitation exercises and thus using them as motivators to aid patient compliance and support personalized treatment [17].

However, it has also been shown that the accuracy and reliability of IMU-based measurements depends on the measured joint and task that is being performed. Additionally, observed accuracies depend on the sensor specifications and software algorithms of the tested IMU [1]. Recent technological advances include higher sampling rates and accuracy of the sensors, as well as lower drift. Therefore, a task and joint specific validation of each new IMU system is required, before such technology can be used routinely in clinical assessments.

The aim of this study is to compare two devices developed for joint angle measurement in clinical applications and rehabilitation: IMU against OMC. In this paper, two commercially available systems were chosen, DyCare^®^ Lynx as IMU device and the Vicon^®^ as OMC system, respectively. Accuracy and usability are important criteria for clinical application, which is why the systems are compared in these two aspects.

## 2. Materials and Methods

### 2.1. Participants

Ten healthy subjects (5 female, 5 male, 172 ± 10.4 cm, 68 ± 13.8 kg, 24.4 ± 2.2 years) with right dominant hands were included in the study. The length of their palm, measured at the third metacarpal bone, ranged from 58 to 86 mm (mean: 70.8 mm; standard deviation (SD): 7.8 mm). All test persons provided informed consent for their data to be used for research purposes, and all their related data were anonymized. Exclusion criteria included any inflammatory joint diseases or other diseases that prevent an exact data acquisition, inadequate language knowledge to follow instructions by the investigative team, legal incompetence, and participation in other medical device studies that could influence this study. This study was conducted in accordance with the Declaration of Helsinki and was carried out with permission from the local ethics committee of the Canton of Zurich, Switzerland (Kek-ZH-Nr: 2018-00457).

### 2.2. Setup

The two IMU were applied with double-sided, hypoallergenic adhesive tape to the skin of the dorsal right hand and forearm of the test subjects, as shown in Figure 1. Skin markers were applied to anatomical landmarks, as described by Reissner et al. [4], except that the markers at the third carpometacarpal joint and at the head of the third metacarpal were not used in this study because they would interfere with the positioning of the distal sensor. The sensor on the hand was placed outside the joint area of the wrist and the metacarpal joint with its longitudinal axis aligned along the metacarpal bone of the middle finger. The proximal sensor was attached to the distal forearm slightly proximal to the wrist joint with its longitudinal axis aligned along the longitudinal axis of the forearm. Care was taken to ensure that the sensors are aligned parallel to each other with their reference points pointing proximally, that a distance of at least 3 cm between the sensors is maintained and that the sensors do not interfere with the subject’s normal motion (particularly during maximum wrist extension). 

For the first method of measurement using the OMC system (OMC1), four reflective markers with diameters of 9 mm were attached to the elbow, four 5 mm markers to the forearm, and five 5 mm markers to the dorsal aspect of the hand. These markers are shown in red in Figure 1. Furthermore, three reflective markers of 5 mm diameter were placed on each IMU for a direct comparison of the two systems, depicted in dark blue in Figure 1. The data acquired using these markers are referred to as OMC2.

### 2.3. Experimental Protocol

Test subjects were assessed on two different test days using the same protocol: First, a static reference trial in neutral position was recorded. Subsequently, five trials of wrist flexion-extension and five trials of radial-ulnar deviation were recorded simultaneously with the IMU and OMC. During each trial, test subjects were instructed to perform 3 cycles of the respective movements. They were also instructed to use the entire range of motion for the movement. The flexion-extension movement was performed with the upper arm fully adducted to the body, the elbow flexed to 90° and the hand in a neutral upright position between pronation and supination. The radial-ulnar deviation movement was carried out with the hand lying flat on the examination table. The whole experiment was then repeated on a different day.

To assess comfort of the wearable IMU, a questionnaire looking into six dimensions of comfort as proposed by Knight and Baber was completed by each test subject. The six dimensions include emotion, attachment, harm, perceived change, movement, and anxiety [18]. The scale ranges from one representing weakest to five representing strongest feelings in each dimension. IMU and OMC markers were placed on all test subjects by the same physician and the required time was measured for both systems. We consider this to be representative of a typical user. All results of the questionnaire and time measurement are presented as mean ± standard deviation.

### 2.4. Technologies Used

The OMC (VICON^®^ MX3+ and VICON^®^ MX3 motion capture system, Oxford Metrics Ltd., UK) used eleven cameras with a resolution of 659 × 493 pixel, recording at a frequency of 100 Hz. Data collection and processing was performed using the corresponding software VICON^®^ Nexus (version 2.3). The IMU (DyCare^®^ Lynx) used two sensors, each consisting of a gyroscope, accelerometer and magnetometer and recording at a sampling frequency of 102.4 Hz. The sensors of the dimensions of 50 mm × 34 mm × 14 mm were configured in the following way: Gyroscope: 2000°/s, Accelerometer: 2 g, Magnetometer 4.7 Ga. Data collection and processing of the IMU system was performed using the software DyCare^®^ Lynx (version 1.7.0) provided by the manufacturer. Trials were recorded selecting the “free joint” setting.

### 2.5. Kinematic Analysis

#### 2.5.1. OMC Data

The post-processing of the recorded data of the OMC system was performed in VICON^®^ Nexus. This included 3D reconstruction of the marker positions from the recordings of the individual calibrated cameras as well as labelling of the markers. No filtering or gap-filling was used. Further data analysis was carried out in Matlab (R2016b). The kinematic evaluation of the OMC was based on marker clusters, the segments were considered as rigid bodies and a least-squares fit was used to determine their rotations [19]. At least three markers per segment had to be visible simultaneously to allow for three-dimensional kinematic motion analysis of the segments. As described by Reissner et al., a functional approach from List et al. [19] was adapted to the wrist to determine joint centers and axes in this study [4]. The wrist joint center was simulated as a ball joint and the functional flexion axis as a hinge joint. Joint movements were calculated according to Grood et al. [20], compliant with the standards established by the International Society of Biomechanics [21]. This analysis was performed for the marker clusters on the skin (OMC1) as well as the markers on the IMU (OMC2). The OMC measurement require the markers to be visible by at least two cameras at any time, otherwise missing values occurred. For such signals containing missing values, the curves were interpolated by a spline interpolation (see Appendix B, Figure A1 for further information).

#### 2.5.2. IMU Data

The kinematic evaluation of the IMU was done by the DyCare^®^ device. No filter was used for processing the IMU signal. The wrist angles calculated by the device were then imported into Matlab for further comparison. For all measurement methods, wrist joint flexion and radial deviation are presented as positive angles whereas extension and ulnar deviation are presented as negative angles.

### 2.6. Data Analysis

The further steps of data evaluation were performed in the same way for the angle curves derived from OMC and IMU. Offset correction between the signals was performed by subtracting the measured wrist angle at the reference position from the whole curve of each dynamic trial for each measurement system. The maxima and minima of the IMU and OMC signals were detected using *findpeaks*, a built-in function of Matlab. The algorithm used is based on detecting zeros in the signal derivative, using approximately 20% of the ROM as minimum peak prominence (5° for the radial-ulnar deviation, 18° for the flexion-extension task). The correct detection of minima and maxima was checked and, if necessary, the minimum peak prominence was increased by 2° to detect only the relevant peaks.

The ROM of each trial was calculated as follows:(1)$$ROM_{ij}={\bar{M}}_{ij}-{\bar{m}}_{ij},$$
whereby *i* represents the subject number, *j* the trial number, k number of maxima ($M_{k}$) and minima ($m_{k}$) identified (within the trial), $M=\left[M_{1},M_{2}\ldots M_{k}\right],\;\bar{M}$ the average of *M*, $m=\left[m_{1},m_{2}\ldots m_{k}\right],$
$\bar{m}$ the average of *m*. Hence, for each trial, the average maxima ($\bar{M}$) and minima $\left(\bar{m}\right)$ were calculated first, then the difference was calculated to determine the ROM of each trial.

The dispersion of the maxima was calculated as follows:(2)$$SDmax_{i}=mean\left(std\left(M\right)\right),$$
whereby *i* represents the subject number, *j* the trial number and ***M*** the array with all the maxima of the trial *j*.

This parameter is an indicator of the consistency of each method detecting maxima and minima, and consequently, the ROM of each trial.

The mean absolute difference (MAD) between ROM among systems was calculated as follows:(3)$$MAD_{ijk}^{ab}=abs(ROM_{ijc}^{a}-ROM_{ijc}^{b}),$$
whereby *i* represents the subject number, *j* the trial number, *c* the cycle number (within the trial) and *a*, *b* the acquisition systems.

Data of both sessions were used to calculate the average ROM for each system as well as the average MAD between the different systems.

### 2.7. Statistical Analysis

The MAD between the different measurement systems was used to demonstrate the accuracy. In addition, the SD within the measurement as well as the difference between testing sessions were calculated to quantify the variability within the measurement and the measurement error for each device, respectively. To assess the statistical significance of the parameters calculated among the three acquisition systems, a Wilcoxon rank sum test (α = 0.05) was utilized. Normality and equal variance of the data was checked before performing the statistical analysis using a Shapiro-Wilk parametric hypothesis test and Bartlett’s test, respectively.

## 3. Results

An overview over the most relevant results is presented in Table 1, indicating coefficient of variation (CV) values of 8.5% ± 3.6% for flexion-extension and 4.9% ± 2.4% for radial-ulnar deviation between the IMU and the OMC1 system.

The mean absolute difference (MAD) in ROM for the flexion-extension trials was 10.4° ± 4.2° and for the radial-ulnar deviation 2.4° ± 1.2° between the IMU and the OMC1 as shown in Table 2. The total ROM for flexion-extension calculated by the IMU was 126.0° ± 18.9°, the OMC1 method measured 136.0° ± 18.0° and the OMC2 method measured 141.4° ± 20.6°, as shown in Figure 2a. The ROM for the radial-ulnar deviation trials was calculated as 50.5° ± 9.0° by the IMU, 51.0° ± 8.4° by the OMC1 method and 49.9° ± 8.4° by the OMC2 method, as shown in Figure 2b. Detailed results with total ROM values in both movement axes for each subject separately can be found in Appendix A, Table A1 and Table A2 and mean SD values of each subject in Table A3 and Table A4.

The signals obtained by the OMC system contained some samples of undefined value due to marker occlusion, which was the case in 3.7% of the samples over the data set of the flexion-extension trials. These artefacts were corrected by recovering the missing values of the signal using a spline interpolation, which was applied to 25% of the flexion-extension trials. However, we found that changing the interpolation method had no significant effect averaged over all subjects. No interpolation was needed in the radial-ulnar deviation trials.

A large overlap between the three distributions can be observed in Figure 3a as well as a slight underestimation of the flexion-extension ROM when IMU measurements are compared to OMC1 and OMC2 assessments. For the ROM of radial-ulnar deviation, a large overlap and similar median values among the three distributions can be observed in Figure 3b.

A Wilcoxon rank sum test was applied to compute the statistical significance of the different ROMs. A threshold of α = 0.05 was used. For the flexion-extension task, a *p*-value of 0.24 and a *p*-value of 0.14 were obtained when the ROM distribution of the IMU acquisition was compared to the one of OMC1 and OMC2, respectively. For the radial-ulnar deviation, *p*-values of 0.62 and 0.85 were obtained when testing the differences in ROM between the IMU versus OMC1 and OMC2 systems, respectively. In both movement axes, the differences between the distributions were above 0.05, indicating that in both cases they are not statistically different.

To check whether a systematic bias is present, Figure 3, Figure 4 and Figure 5 show Bland-Altman plots comparing the IMU system’s ROM, the OMC 1 ROM and the OMC2 ROM respectively, each for both flexion-extension and radial-ulnar deviation movements. Normal distribution of the differences was confirmed with the Shapiro-Wilk parametric hypothesis test using a threshold of α = 0.05.

A systematic difference of −10° between ROM of the IMU compared to ROM of the OMC1 method and a systematic difference of −15.5° when comparing the ROM of the IMU with the OMC2 method for flexion-extension movements was found. Between the measurements with the two different marker sets (OMC1 and OMC2), there was a systematic difference of −5.5°. There was no systematic difference between the different methods for radial-ulnar deviation movements.

The questionnaire investigating wearing comfort of the IMU system resulted in values of 1.3 ± 0.64 for emotion, 1.8 ± 0.87 for attachment, 1.3 ± 0.46 for harm, 1.1 ± 0.3 for perceived change, 1.3 ±0.46 for movement, and 1 ± 0 for anxiety, 1 representing weakest and 5 strongest feelings in the respective dimension. The IMU positioning took 36 ± 33 s and the marker placement 313 ± 64 s.

The average SD value of the minima and maxima in the three cycles was <2.2° within the trial and <2.7° within the session (Table 3 and Table 4). The absolute values of the individual test-retest differences ranged between 0.0° and 24.4° (Table 5). In some test participants, we observed large test-retest differences (absolute difference >5°) for all measurement systems (e.g., S5–S7 for flexion-extension, S6 for radial-ulnar deviation), in other participants the large differences were only present in certain methods (e.g., S1–S2 for flexion-extension).

## 4. Discussion

This study investigated the use of an IMU (DyCare^®^) for three-dimensional motion analysis of the wrist compared to an OMC system (Vicon^®^), which represents the current laboratory gold standard. The tested IMU has shown an excellent accuracy regarding the radial-ulnar deviation (2.3° MAD). The higher ROM in flexion-extension puts its higher mean absolute difference (MAD 10.3°) in perspective (8%). Furthermore, a systematic underestimation of the ROM of 10° (8%) and 15.5° (11%) by the IMU compared to the OMC1 and OMC2 methods, respectively, was found in this movement axis. The systematic difference in ROM of radial-ulnar deviation movements was close to 0° (1%). It seems that the agreement between the two measurement systems depends on the magnitude of the measured ROM, with smaller ROM leading to more reliable results. This might be due to the anatomical model used by the manufacturers evaluation software, which presents a known problem when comparing an IMU based system with an OMC system [16,22]. There are three different options to define the local coordinate system, whereby each might lead to a different alignment and thus affect the resulting joint angles: anatomical frames based on bony landmarks, reference frames aligned with the global coordinate system or functional frames determined by the relative motion of the segments [23]. Another possible explanation is that the measurement accuracy of the IMU system might be depending on the absolute three-dimensional orientation of the sensors in the room, as the magnetometer is depending on the magnetic north and the accelerometer on gravity. Furthermore, velocity of the recorded motion may also affect the IMU accuracy [24].

There was also a systematic difference of 5.5° (4%) when comparing the results of the two different marker sets (OMC1 and OMC2). The distribution of the markers influences the measurement with an OMC system and maximizing the mean marker cluster radius is considered preferable [25]. Unexpectedly, we found greater differences when comparing the IMU with the markers on the sensors themselves (OMC2), than with the markers in anatomical positions on the skin (OMC1). The underestimation of the joint angles by the IMU might to some extent have been compensated by the sensors partially lifting off from the underlying bone in end-range motion (which by itself would lead to an overestimation of the joint angle). Such an effect has been visually observed by the investigators, especially in small hands, but cannot be quantified on the basis of the available measurement data.

The wearing questionnaire’s results indicate excellent acceptance of the IMU by the test subjects. The preparation of the test subjects took almost nine times longer for the OMC system compared to the IMU, a difference which is further enhanced by the time-consuming post-processing required for the recordings of the OMC system.

The observed differences between the IMU and the OMC1 as a reference system of 10.3° ± 4.6° for flexion-extension and 2.3° ± 1.3° for radial-ulnar deviation are even smaller compared to previous studies investigating the wrist. One study found differences of 6° radial-ulnar deviation and 14° for flexion-extension movements [16] while another study comparing different calibration methods for an IMU system found MAD values between IMU and OMC systems ranging from 9.6° to 24.1° for flexion-extension and from 10.8° to 17° for radial-ulnar deviation [11]. Perez et al. even found MAD values of 26.9° for wrist flexion-extension, but like the previously mentioned study, they investigated movements of the whole upper limb [12], which could affect the wrist measurement in contrast to a wrist-specific setup. Our results are more similar to gait analysis studies, where differences from 3° to 11° between OMC and IMU based systems were found [14]. Compared to an earlier study comparing goniometer and OMC [4], the ROM measurements using the IMU are more accurate than the goniometer (especially for the radial-ulnar deviation). Therefore, both examined devices outperform the currently accepted clinical measurement tool.

We observed small measurement errors (average SD < 2.6°) within the session, indicating the potential of the method. However, compared to a previous study, the test-retest agreement for the flexion-extension ROM (MAD 7.7–9.4°, Table 5) is considerably lower [4]. The presence of the sensor and the differences in the marker set may have affected measurement repeatability. Furthermore, the first study focused on repeatability [4] whereas this study focused on the comparison of different measurement systems. The instruction of the subjects could be relevant for the repeatability of the measurements. In addition, when looking at the individual values, large test-retest differences for all measurement systems suggest that these volunteers may have actually performed a different movement in the first and second measurement. If, in contrast, the large individual differences are only present in certain methods, this is more likely to indicate a measurement-associated error, such as poor alignment of the joint axes. However, to assess the underlying cause of the error, validation with an imaging technique would be required, which goes beyond the scope of this article.

Occlusion of markers presented a difficulty with the camera-based method (in the flexion-extension task) and resulted in the need to reconstruct a relevant portion of the data acquired by the OMC using interpolation. On the other hand, the IMU measurement resulted in a continuous recording of the joint angle without the need of data post-processing. This represents a significant advantage of the IMU, especially with regard to the measurement of more complex tasks, such as daily activities, during which the visibility of the markers is even more challenging. Although the OMC is considered the gold standard for motion analysis, it is also affected by measurement error, mainly caused by skin movement artifacts and possible misalignment of the joint coordinate system [10,26]. To assess absolute accuracy, a comparison to an imaging technique would be required. However, for evaluative tools that aim to measure change in clinical health status, the repeatability is a core factor [27]. It would certainly be interesting to test the DyCare^®^ system regarding its inter-observer variability as it has been carried out for other IMU based systems with good results [11].

The IMU based system has shown excellent accuracy in radial-ulnar deviation movements. Therefore, it seems highly suitable to be applied to research involving this movement axis. The presence of a systematic bias when comparing the flexion-extension ROM of IMU with the two OMC based methods points to the possibility for adaptation of the IMU measurement, which could then result in improved agreement between the two measurement systems for these movements. Whether this systematic difference was caused by a systematic underestimation of the angle by the sensor itself, or by factors specific to the application in motion analysis, such as a different orientation of the joint axes, should be further investigated. If wrist-specific differences are present, a possible solution is the use of the same anatomical model for both the IMU and the OMC based systems [16,22]. The functional determination of the joint coordinate system, as it is performed in our kinematic model for the evaluation of the OMC data, is considered to increase measurement repeatability and interpretability, reduce the influence of marker/sensor placement and reduce kinematic crosstalk [28]. We therefore suggest considering the implementation of a functional approach for the IMU as well. Another possibility would be the alignment of the coordinate systems relative to specific anatomical landmarks. The software of the IMU also offers the option to align the movement IMU (distal sensor) with the reference IMU (proximal sensor) by multiplying the conjugated quaternion that describes their relative position. However, such an approach is still dependent on the placement of the proximal sensor and its application so far is uncommon in motion analysis. The effects of the proposed changes in the definition of joint coordinate systems require further investigation, in particular its impact on the magnitude of kinematic crosstalk. However, it should also be acknowledged that part of the observed difference between the systems might be caused by the missing values due to marker occlusion in the OMC system, as they occurred in critical positions in the angle curves and caused the need for interpolation of the curves.

According to Cuesta-Vargas, depending on the anatomical area investigated, the sensor size is also an important consideration [1]. We found the sensors to be relatively large (50 mm) in relation to the length of the hand (length of third metacarpal ranges from 58–86 mm). Additionally, the palm segment, which consists of several individual bones, was observed to change shape during movement and has a certain curvature, especially in the maximum flexion position. This effect is due to flexion-extension movements up to 20° and 27° in the fourth and fifth carpometacarpal joints [29]. For small hands, it was difficult to place the sensors in such a way that neither the wrist nor the MCP joints are restricted in their freedom of movement. Smaller sensors or an intermediate piece between the sensor and the hand with a smaller contact surface would be possible solutions.

The accuracy of the IMU may be improved by performing tasks with the sensors moving in the vertical plane. A comparison of the same tasks in different orientations was not performed in this study but could be useful to determine the optimal motion execution for ROM measurements with the IMU system. Depending on the task, however, it is not possible or useful to change the orientation of the movement e.g., for the investigation of activities of daily living. The advantage of the OMC systems is that the accuracy is independent of the orientation of the hand in the room.

The IMU system shows excellent accuracy (2.3° MAD) with radial-ulnar deviation and a fair accuracy of about 10° with flexion-extension movements with even better accuracy below 140°. MAD values of up to 5° are often regarded as reasonable for clinical application [1], while other authors regard differences up to 10° as reasonable for occupational biomechanics application [16]. The obtained results are within these margins and we expect even smaller errors when investigating wrist pathologies and in postoperative outcome studies, where patients have a reduced total ROM. However, an investigation of the IMU during more complex activities of daily life, in addition to the standardized measurements performed in this study, would complement the assessment of the clinical applicability. The problem of marker occlusion and extensive post processing is partly responsible for the absence of OMC based systems in a clinical setting, whereas an IMU based system does not suffer from these limitations.

Usually joint mobility is measured with the adjacent joints in neutral position. The movement tasks suggested by the sensor manufacturer have been slightly adapted in this study in order to improve marker visibility as well as to ensure a neutral position in the radioulnar joint. This must be taken into account when interpreting the results.

The evaluation of data from OMC systems requires advanced knowledge about motion analysis, as there are no standard routines available. The integrated data evaluation of the DyCare^®^ Lynx software is therefore a major advantage of the IMU with regard to the applicability of motion analysis in clinical practice. Measurements with IMU are much more efficient as no data post-processing is required. In addition, the ROM values obtained can also be monitored in real-time, whereas the evaluation of the OMC data usually takes place after completion of the data acquisition and in the absence of the patient. It may be useful for the treating surgeon to obtain the ROM values already during the examination, as joint mobility is an important factor for clinical decision-making.

## 5. Conclusions

This study highlights the potential of the IMU based DyCare^®^ Lynx system for wrist motion analysis. Their setup allows for accurate, effortless kinematic motion analysis with low intra-observer variability. The results of the investigated IMU system are even more precise in comparison to similar studies. In the current setup, there is a systematic underestimation of movement in the flexion-extension axis by the IMU relative to the OMC. An alignment between both acquisition systems could further improve the results. In particular, the implementation of a functional joint axis determination or the effect of a mathematical alignment of the sensors in the reference position should be examined. Still, IMU based systems are, based on our results, within the generally acceptable margin of error for joint angle measurement of the wrist and more accurate than goniometer measurement, which is the current clinical standard. However, whether the observed accuracy is appropriate can only be determined in the context of the intended application.

## Figures and Tables

**Figure 1 sensors-19-05297-f001:**
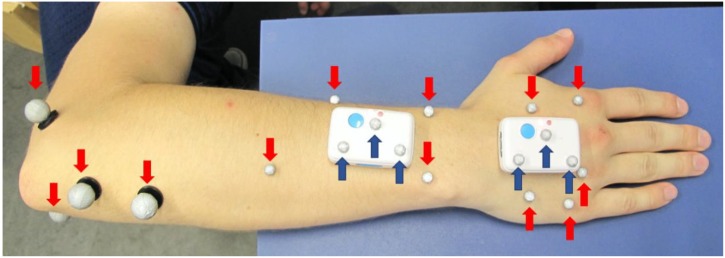
Sensor (IMU), skin marker (red arrows, OMC1) and sensor marker (dark blue arrows, OMC2) placement.

**Figure 2 sensors-19-05297-f002:**
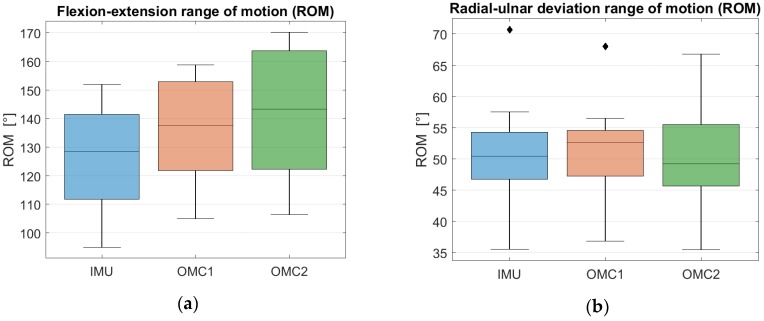
(**a**) Flexion-Extension average ROM values for IMU, OMC1 and OMC2 acquisition systems. (**b**) Radial-ulnar deviation average ROM values for IMU, OMC1 and OMC2 acquisition systems.

**Figure 3 sensors-19-05297-f003:**
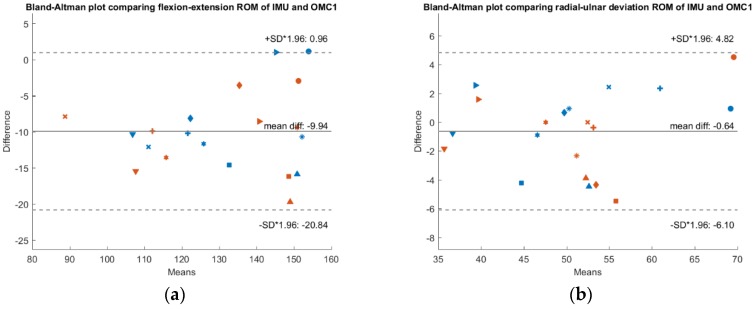
Bland-Altman plot comparing IMU ROM and OMC1 ROM. The x-axis depicts the mean ROM of both systems, while y-axis shows their difference. A confidence interval of ±1.96 * SD is plotted in dashed lines. Results of measurement day 1 are depicted in blue and measurement day 2 in red. Each subject is represented by a different marker (**a**) Results of the flexion-extension trials. (**b**) Results of the radial-ulnar deviation trials.

**Figure 4 sensors-19-05297-f004:**
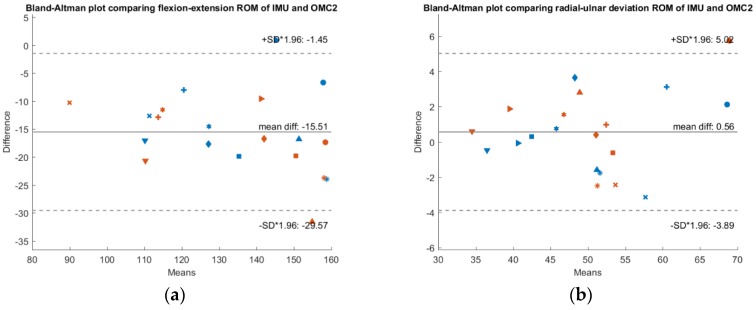
Bland-Altman plot comparing IMU ROM and OMC2 ROM. The x-axis depicts the mean ROM of both systems, while y-axis shows their difference. A confidence interval of ±1.96 * SD is plotted in dashed lines. Results of measurement day 1 are depicted in blue and measurement day 2 in red. Each subject is represented by a different marker (**a**) Results of the flexion-extension trials. (**b**) Results of the radial-ulnar deviation trials.

**Figure 5 sensors-19-05297-f005:**
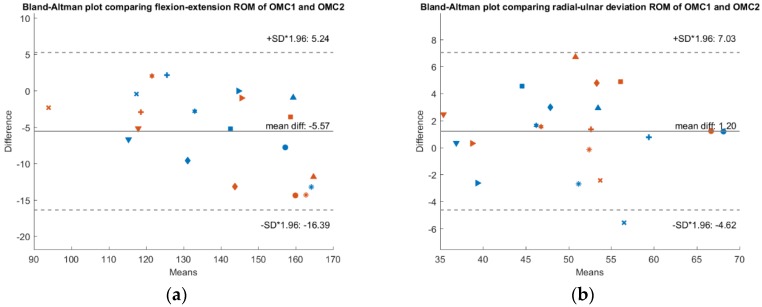
Bland-Altman plot comparing OMC1 ROM and OMC2 ROM. The x-axis depicts the mean ROM of both systems, while y-axis shows their difference. A confidence interval of ±1.96 * SD is plotted in dashed lines. Results of measurement day 1 are depicted as blue and measurement day 2 as red. Each subject is represented by a different marker (**a**) Results of the flexion-extension trials. (**b**) Results of the radial-ulnar deviation trials.

**Table 1 sensors-19-05297-t001:** Summary of the main results. Results in percentage represent the coefficient of variation (CV) using the first element of difference as a reference.

	ROM IMU (°)	ROM OMC1 (°)	ROM OMC2 (°)	MAD IMU-OMC1 (°)	MAD IMU-OMC2 (°)	MAD OMC1-OMC2 (°)	CV IMU-OMC1 (%)	CV IMU-OMC2 (%)	CV OMC1-OMC2 (%)
Flexion-extension	126.01 ± 18.85	135.96 ± 17.98	141.35 ± 20.64	10.37 ± 4.22	15.48 ± 5.55	6.21 ± 4.18	8.55 ± 3.59	12.42 ± 4.20	4.76 ± 2.85
Radial-ulnar deviation	50.48 ± 9.04	51.05 ± 8.44	49.91 ± 8.44	2.42 ± 1.19	2.11 ± 0.85	2.67 ± 1.41	4.86 ± 2.41	4.06 ± 1.06	5.42± 2.80

**Table 2 sensors-19-05297-t002:** Average MAD ROM values between IMU and OMC1, IMU and OMC2 and OMC1 and OMC2 for flexion-extension and radial-ulnar deviation.

Flexion-Extension	IMU-OMC1 (°)	IMU-OMC2 (°)	OMC1-OMC2 (°)	Radial-Ulnar Deviation	IMU-OMC1 (°)	IMU-OMC2 (°)	OMC1-OMC2 (°)
S1	3.04	12.00	11.10	S1	2.87	3.94	1.26
S2	5.07	5.66	1.46	S2	2.07	1.49	1.77
S3	10.04	10.43	2.76	S3	1.57	2.08	1.16
S4	10.08	23.85	13.77	S4	1.50	2.06	2.13
S5	10.02	11.42	1.85	S5	1.70	2.80	4.01
S6	15.40	19.84	4.45	S6	4.86	1.21	4.70
S7	7.20	17.34	10.93	S7	2.80	2.58	3.87
S8	17.36	22.36	7.27	S8	4.19	2.48	4.80
S9	12.89	18.85	5.96	S9	1.38	0.82	1.42
S10	12.63	13.03	2.56	S10	1.25	1.68	1.63
Mean	10.37	15.48	6.21	Mean	2.42	2.11	2.67
SD	4.22	5.55	4.18	SD	1.19	0.85	1.41

**Table 3 sensors-19-05297-t003:** Average standard deviation of the maxima (MAX, corresponding to wrist flexion or radial deviation) and minima (MIN, corresponding to wrist extension or ulnar deviation) within each trial.

Task	SD Max IMU (°)	SD Min IMU (°)	SD Max OMC1 (°)	SD Min OMC1 (°)	SD Max OMC2 (°)	SD Min OMC2 (°)
Flexion-Extension (mean)	2.20	2.17	1.78	2.10	1.89	2.20
Flexion-Extension (SD)	0.84	0.49	1.01	0.42	1.04	0.54
Radial-Ulnar (Mean)	1.07	1.42	0.76	1.14	0.63	1.22
Radial-Ulnar (SD)	0.36	0.75	0.30	0.95	0.27	0.89

**Table 4 sensors-19-05297-t004:** Average standard deviation of the maxima (MAX, corresponding to wrist flexion or radial deviation) and minima (MIN, corresponding to wrist extension or ulnar deviation) within each session.

Task	SD Max IMU (°)	SD Min IMU (°)	SD Max OMC1 (°)	SD Min OMC1 (°)	SD Max OMC2 (°)	SD Min OMC2 (°)
Flexion-Extension (Mean)	2.69	2.42	2.16	1.79	2.47	2.12
Flexion-Extension (SD)	1.55	1.50	1.49	1.31	1.83	1.35
Radial-Ulnar (Mean)	0.77	1.05	0.58	1.01	0.49	1.07
Radial-Ulnar (SD)	0.47	0.60	0.35	0.81	0.23	0.75

**Table 5 sensors-19-05297-t005:** Test-Retest Difference of the ROM within each system (IMU, OMC1 and OMC2) for flexion-extension and radial-ulnar deviation.

Flexion-Extension	IMU (°)	OMC1 (°)	OMC2 (°)	Radial-Ulnar Deviation	IMU (°)	OMC1 (°)	OMC2 (°)
S1	4.7	0.6	−6.0	S1	−2.2	1.4	1.5
S2	9.2	−0.3	−1.3	S2	0.1	−0.8	2.1
S3	9.2	9.5	4.4	S3	9.2	6.4	7.0
S4	0.6	1.9	0.9	S4	0.8	−2.5	0.0
S5	20.2	24.4	22.6	S5	3.7	1.2	4.4
S6	−15.2	−16.8	−15.2	S6	−10.4	−11.7	−11.3
S7	−15.0	−9.8	−14.2	S7	−1.2	−6.3	−4.5
S8	6.9	1.7	−11.8	S8	0.1	0.7	4.5
S9	1.8	−3.4	−1.9	S9	1.5	0.4	2.5
S10	10.9	9.0	13.9	S10	−1.2	−0.5	−0.4
Mean	3.3	1.7	−0.9	Mean	0.0	−1.2	0.6
SD	10.6	10.7	11.5	SD	4.7	4.6	5.0
MAD	9.4	7.7	9.2	MAD	3.0	3.2	3.8

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
