# Peer review of "Comparison of a New Inertial Sensor Based System with an Optoelectronic Motion Capture System for Motion Analysis of Healthy Human Wrist Joints"

_sensors, 2019, doi:10.3390/s19235297_

Round 1

Reviewer 1 Report

This manuscript tested concurrent validity of commercial IMU in comparison to motion capture system. In general, this manuscript presented the study quite well. I just have a few comments that the authors might consider to improve the manuscript.

Lines 45-54: The authors seem to understand motion capture is considered to be accurate (and this was stated in the abstract). But then the authors criticized the accuracy of motion capture due to skin artifact and so on. I think the last sentence in this paragraph is conflicting and does not really help the authors rational so I suggest remove the last sentence in this paragraph.

Lines 55-68: I am not sure why IMU needs to be further validated in this wrist movement especially there are some papers for the upper extremity. I understand the authors argued that task-specific validity needs to be tested but the paper was published almost 10 years ago. Please provide why previous validation papers (especially upper extremity papers) did not provide enough rationale to use IMU for wrist flex-ext or radial-ulnar deviation.

Line 95: patient? Were the subject patient population?

Figure 1: Please use an actual photo instead of drawing because it will help readers better understand the experimental procedures.

Lines 125-133: Were the data filtered somehow? If so, what kind of filter(s) was used and what was the cutoff frequency?

Throughout the manuscript, please be consistent with IMU, IMUs, IMU sensors (just choose one of them, and keep consistent).

Also, throughout the manuscript, please double check with some minor language mistakes (e.g., singular, plural.)

Reviewer 2 Report

This study aims to validate the use of an IMU-based system for wrist-motion assessment against an optoelectronic motion-capture system. The motivation behind this validation regards the potential use of this IMU-based system in the clinic. I like the validation approach taken by the authors that includes usability criteria. However, I have concerns regarding the objectives and organization of the manuscript which I would appreciate to clarify prior to publication.

Introduction:

The authors make a good introduction of the potential and limitations of both the OMC and the goniometer. They also introduce the inertial technology (advantage, possibilities and importance to validate for specific tasks), but not much is said on the inclusion of the technology within a clinical tool (limited possibilities at the moment specifically for application such as wrist assessment). Yet, the study looks into usability criteria (time for setup, acceptability). I think these aspects should be announced in the introduction. IMU details: Line 56: please revise sentence (and/or magnetometer). Accelerometers, gyroscopes and magnetometers are all considered inertial sensors. However, when we refer to IMU for biomechanics, it is usually accelero, accelero/gyro, or accelero/gyro/magneto. Line 71: Results may also vary according to the specificity of the inertial system (either the sensors specifications or the software algorithms). Alignment (Line 74-77): The authors are right that different alignment and models are used and that has some effect on the data (expressed in anatomical reference). Yet, this sentence appears to announce that the study will evaluate the differences between the different approaches which is not the case. Please revise. Aim: The aim stated by the author is very large. The introduction led me to believe that consideration on the accuracy relative to a clinical use will be performed which is not the case (only comparison between IMU and OMC). Yet, usability criteria were also evaluated. I suggest revising the wording of the aim and adding specific sub-objectives. My vision: Lynx is an IMU-based system developed for clinical use (rehab). Validation therefore includes accuracy criteria, and usability. This study looks at both.

Methods:

Sample size: any sample size justification? Impact of n on the results? OMC markers placement: I originally had a question about the choice of the markers’position. I then saw that you included this info in the figure caption. I think it should be in the main body… Information on the protocol is available, yet I had to go back, after reading the results, to make sure all info was there. I suggest clarifying the content: Participants were assessed twice (two sessions, different days) => Each session is composed of 5 trials of wrist FE, 5 trials radial-ulnar dev. => Each trial, participant was instructed to… for 3 cycles. Lines 116-117: Variability within the measurements… I believe this goes in the section “statistical analysis” Please clarify: all data were used regardless of session, cycle…? Variations within participants? Between sessions? Time to position the markers and the IMU: Please specify who did the experiment. Is it representative of a typical targeted user? (i.e. rehab professional) Variable IMU: Please specify what comes out of the Dycare Lynx… Is it directly the ROM or orientation of both sensors and computation was made in post-processing? OMC variable: data was post-processed… Details? Filtering used? What are the estimated impacts? Interpolation: Although this validation gives an idea of the rigorous process followed by the author, I suggest removing the specific results regarding this aspect as it is not directly related to the study aim. It could be included in an annex. ROM calculation: Please ensure consistency in the subscripts used. Sometimes, participant id is k, others it’s i… This will make this section a lot clearer… Statistical analysis: I suggest have a section (or a sub-section) entitled “statistical analysis”. This section should include all tests done to answer the objectives… (Variability => SD, accuracy => MAD…) Student’s t-test: I am not sure about the choice of the Student’s t-test because of the sample size. Also, t-test requires independence between observations which does not appear to be the case. If I understand correctly, the authors used all data repetitions, regardless of the session, trial… Bland-altman: I prefer this representation. However, conclusions can only be drawn if the difference follows a normal distribution. Was this assumption met? Results: I am concerned about the variation between participants… The authors chose to performed the test-retest on the ROM values for each participant. The difference therefore includes not only potential variations by the system, but also normal intra-individual variation. Hence, would a test-retest on the system precision give more information? Or an ANOVA considering between/within participants variations?

Discussion:

While the introduction talked a lot about the goniometer (currently accepted clinical measurement tool), nothing is said about it in the discussion. I understand that ROM was not measurement with the gonio throughout the study. However, what can you conclude about the reported accuracy for the clinic? Are the accuracy values enough to allow clinical diagnostic? Rehab progression assessment? Potential source of errors in IMU: velocity of motion may also affect the data (see https://journals.plos.org/plosone/article?id=10.1371/journal.pone.0079945)
